# The risk factors for mortality of diabetic patients with severe COVID-19: A retrospective study of 167 severe COVID-19 cases in Wuhan

Yan Hui[☯], Yi Li[☯], Xiwen Tong[☯], Zhiqiong Wang[☯], Xia Mao, Lifang Huang*, Donghua Zhang[ORCID]*

Department of Hematology, Tongji Hospital, Tongji Medical College, Huazhong University of Science and Technology, Wuhan, Hubei, China

☯ These authors contributed equally to this work.
* zdh_62@126.com (DZ); huanglifang627@163.com (LH)

## Abstract

Diabetes is one of the most common comorbidities in adult patients with coronavirus disease 2019 (COVID-19). This study aimed to analyze the mortality risk factors of diabetic patients with COVID-19. A total of 167 patients with severe COVID-19, including 55 diabetic patients and 112 nondiabetic patients at Tongji Hospital, Wuhan, China from January 28, 2020, to March 10, 2020, were collected. The laboratory, radiological, management information, and medical history was retrospectively reviewed. Potential mortality risk factors in diabetic patients with COVID-19 were evaluated by the proportional hazard Cox model. The clinical information of 167 patients with severe COVID-19 was analyzed. The median age was 65.0 years. Approximately 32.9% of patients had diabetes. In total patients, older age, diabetes, and lymphocyte count were associated with increased risk of death. In diabetic patients, increased mortality was associated with decreased lymphocyte count ($\leq$0.45×10⁹/L, HR 0.196, 95% CI 0.049–0.781, $P$ = 0.021), lactate dehydrogenase >600 U/L (HR 8.010, 95% CI 1.540–41.670, $P$ = 0.013), hsCRP >90 mg/L (HR 4.551, 95% CI 1.472–14.070, $P$ = 0.009) and interleukin-10 >10 U/mL (HR 5.362, 95% CI 1.239–23.199, $P$ = 0.025). COVID-19 patients with diabetes had a poor prognosis, especially when they had two or more of the following abnormalities ($\chi2$ = 58.62, $P$<0.001): lymphocyte count was $\leq$0.45×10⁹/L, lactate dehydrogenase was >600 U/L, hsCRP was >90 mg/L and IL-10 was >10 U/mL. For diabetic patients with COVID-19, more attention should be paid to the dynamic monitoring of cytokine levels, and the control of hyperglycemia.

## Introduction

In December 2019, the outbreak of coronavirus disease 2019 (COVID-19) caused by severe acute respiratory syndrome coronavirus 2 (SARS-CoV-2) first occurred in Wuhan, China, and quickly spread abroad. The clinical manifestations of SARS-CoV-2 closely resembled those of

**Data Availability Statement:** All relevant data are within the manuscript and its Supporting information files.

**Funding:** This work was supported by the National Natural Science Foundation of China (No.81500082). The funders had no role in study design, data collection and analysis, decision to publish, or preparation of the manuscript.

**Competing interests:** The authors have declared that no competing interests exist.

SARS, and infected patients may develop acute respiratory distress syndrome (ARDS) with a high likelihood of admission to the intensive care unit (ICU) and subsequent mortality. In these severe cases, comorbidities are common, especially diabetes [1–3].

Information about the details of clinical course of COVID-19 patients with diabetes has not yet been well described. Several published studies reported the risk factors associated with severity and mortality in patients with COVID-19 [1–3]. To our knowledge, no previous study has been performed on the definite outcomes in patients with diabetes.

Therefore, we aimed to compare 55 COVID-19 patients with diabetes and 112 patients without diabetes to summarize the differences in clinical and laboratory features. In addition, univariate and multivariate analyses were used to evaluate the potential risk factors associated with the mortality of diabetic patients with COVID-19.

## Materials and methods

### Patients

A total of 167 patients with severe COVID-19, including 55 diabetic patients and 112 nondiabetic patients at Tongji Hospital, Tongji Medical College, Huazhong University of Science and Technology, Wuhan, China from January 28, 2020, to March 10, 2020, were retrospectively reviewed. Patients were classified as severe when meeting any one of the following according to the National Recommendations for the Diagnosis and Treatment of COVID-19 (7th edition) [4]: (1) respiratory distress: RR $>$ = 30times/min; (2) fingertip oxygen saturation $<$ = 93% under resting state; (3) arterial partial pressure of oxygen (PaO2)/fraction of inspiration O2 (FiO2) $<$ = 300 mmHg (1 mmHg = 0.133 kPa). The criteria of ARDS was also referred to define severe COVID-19 patients [5]. Tongji Hospital is one of the hospitals responsible for the central treatment of severe SARS-CoV-2-infected adult patients assigned by the government. All patients enrolled in the present study were diagnosed based on the National Recommendations for the Diagnosis and Treatment of COVID-19 (7th edition) [4] and the World Health Organization (WHO) interim guidelines [6]. Poorly controlled hyperglycemia was defined when two or more blood glucose $>$ 10 mmol/L occurred within any 24-hour period based on published criteria, otherwise well controlled hyperglycemia was considered [7]. The present study was approved by the Ethics Committee of Tongji Hospital, Tongji Medical College, Huazhong University of Science and Technology and was consistent with the Declaration of Helsinki. Written informed consent was not obtained from the patients, as the Ethics Committee approved the application for exemption from obtaining informed consent.

### Data collection

The laboratory, radiological, management information and medical history, including the comorbidities, symptoms and signs of patients, were obtained from the electronic medical records from March 11, 2020 to March 18, 2020. All data were reviewed in detail by two experienced physicians (YL and LH). In addition, the dynamic changes in the results of the laboratory tests during hospitalization were monitored.

### Laboratory testing for SARS-CoV-2

Throat swab samples were collected to extract SARS-CoV-2 ribonucleic acid (RNA) from patients suspected to have SARS-CoV-2 infection. The laboratory confirmation of SARS-CoV-2 was achieved through the concerted efforts of the Chinese Center for Disease Prevention and Control (CDC) and Wuhan Institute of Virology. The real-time reverse transcription-

polymerase chain reaction (RT-PCR) assay and diagnostic criteria were performed in accordance with the protocol established by the WHO [6].

## Statistical analysis

Continuous variables were expressed as the median and interquartile range (IQR) values. Categorical variables were described as counts and percentages. Continuous variables were compared using independent group t-tests when the data were normally distributed. Otherwise, the Mann-Whitney test was used. Categorical variables were compared by chi-square test or Fisher's exact test. The proportional hazard Cox model was used, and univariate models were fitted with a single candidate variable one at a time. We used X-tile software to identify cutoff points for evaluating potential mortality risk factors in diabetic patients with COVID-19 [8]. The subdistribution hazards ratio (SDHR) with a 95% confidence interval (95% CI) was reported. The overall survival (OS) was estimated using the Kaplan-Meier method and compared using the log-rank test. These statistical analyses were performed using SPSS software version 25.0. Statistical significance was defined as a two-sided P-value of less than 0.05.

## Results

We included 167 inpatients with COVID-19 in the final analysis. A total of 109 patients died during hospitalization, and 58 patients were discharged. The median age of the 167 patients was 65.0 years, ranging from 26.0 years to 92.0 years, and an approximately 1.9:1.0 male-to-female ratio was found. Comorbidities were present in most patients, with hypertension (40.1%) being the most common comorbidity, followed by diabetes (32.9%) and coronary heart disease (19.2%). The most common symptoms were fever (88.6%) and cough (70.1%), followed by dyspnea (45.5%) and diarrhea (18.6%). All patients had bilateral infiltrates on the chest X-ray. The treatment information is shown in S1 Table. The median time from illness onset to discharge was 26.0 days (IQR 19.0–31.0), whereas the median time to death was 24.0 days (IQR 18.0–29.0) (S1 Table).

　　We compared the laboratory markers obtained at admission and discharge or death between survivors and nonsurvivors. Levels of interleukin-2 receptor (IL-2R), IL-6, hypersensitive C-reactive protein (hsCRP), lactate dehydrogenase (LDH) and serum ferritin were clearly elevated in nonsurvivors compared with survivors throughout the clinical course and increased as illness deterioration progressed. Levels of lymphocytes at admission declined clearly in nonsurvivors compared with survivors and continued to decrease until the terminal stage was reached (Fig 1, S3 Table). A comparison of these markers performed between patients with and without diabetes who died/survived were showed in S4 Table.

　　In the univariate analysis, the odds of in-hospital death were higher in patients with diabetes than in patients without diabetes. Older age, lymphopenia, leukocytosis, neutrophilia, thrombocytopenia, hyperglycemia and elevated d-dimer, LDH, hsCRP, IL-8, and IL-10 were also associated with death. We included all the 167 patients for all variables in the multivariate Cox regression model. We found that older age, diabetes, and hyperglycemia, lymphopenia and neutrophilia at admission were associated with increased risk of death (S2 Table).

　　The median age of diabetic patients was greater than nondiabetic patients (P = 0.001). We compared the laboratory results on admission between diabetic patients (n = 55) and nondiabetic patients (n = 112). Compared with nondiabetic patients, diabetic patients had significantly higher leucocyte count (P = 0.005), neutrophil count (P = 0.004), levels of d-dimer (P = 0.018), LDH (P = 0.007), hs-cTnT (P<0.001), BNP (P = 0.032), procalcitonin (P = 0.005), serum ferritin (P = 0.002), IL-6 (P = 0.001), IL-10 (P = 0.001) and TNF (P = 0.014). Diabetic patients had lower albumin levels than nondiabetic patients (P<0.001). Increased levels of IL-

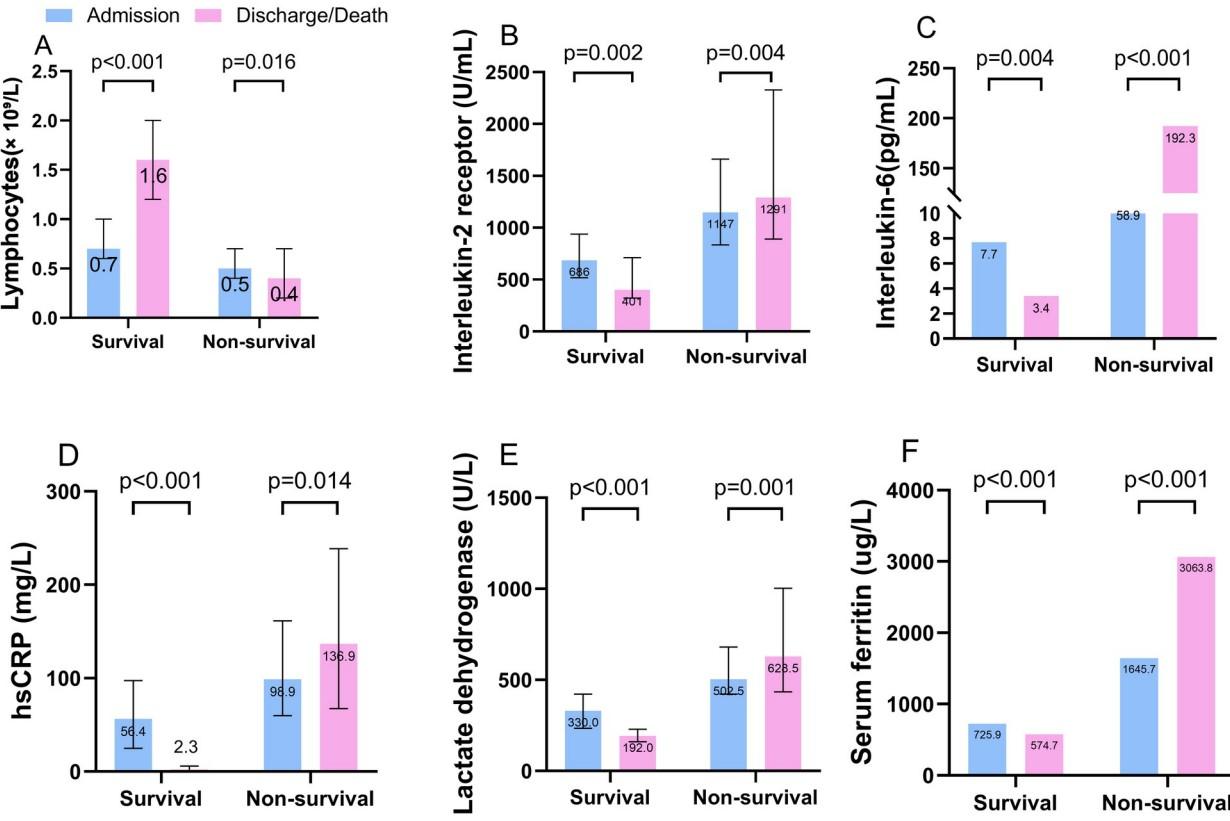

**Fig 1. Laboratory markers obtained between admission and at discharge or death among survivors and nonsurvivors.** The figure shows the temporal changes in lymphocytes (A), interleukin-2 receptor (B), interleukin-6 (C), hsCRP (D), lactate dehydrogenase (E), and serum ferritin (F). The differences between survivors and nonsurvivors were significant at admission and at discharge or death. hsCRP = hypersensitive C-reactive protein.

2R at admission were observed in the majority of patients, but there was no significant difference between diabetic patients and nondiabetic patients (*P* = 0.053) (Table 1).

We compared the laboratory results on admission between nonsurvivors and survivors in diabetic patients and non-diabetic patients. In diabetic patients (n = 55), compared with survivors (n = 11), nonsurvivors (n = 44) had significantly increased leucocyte count (*P*<0.001), neutrophil count (*P*<0.001), levels of d-dimer (*P*<0.001), LDH (*P*<0.001), hs-cTnT (*P*<0.001), BNP (*P* = 0.001), procalcitonin (*P* = 0.006), serum ferritin (*P* = 0.044), IL-2R (*P* = 0.023), IL-6 (*P* = 0.006) and IL-10 (*P* = 0.023). Nonsurvivors with diabetes had lower lymphocyte count than survivors (*P* = 0.006). Although blood glucose levels in both surviving and nonsurviving diabetic patients were greater than normal, there was no significant difference between the two groups (*P* = 0.265). However, nonsurviving diabetic patients had more poorly controlled hyperglycemia than surviving diabetic patients (*P* = 0.019). Notably, among the nondiabetic patients, the blood glucose of the nonsurvivors was significantly higher than that of the survivors (*P*<0.001). In nondiabetic patients (n = 112), compared with survivors (n = 47), nonsurvivors (n = 65) had significantly decreased level of albumin (*P*<0.001) and increased levels of IL-8 (*P*<0.001) and TNF-α (*P* = 0.045) (Table 2).

In the univariate analysis, the odds of in-hospital death were higher in diabetic patients with lymphopenia than in diabetic patients without lymphopenia. Older age, neutrophilia, thrombocytopenia, hyperglycemia and elevated LDH, serum creatinine, BUN, hs-cTnT, BNP,

**Table 1. Demographic, laboratory findings and treatment of patients on admission.**

| Characteristics | Total (n = 167) | Diabetic patient (n = 55) | Non-diabetic patient (n = 112) | *P* Value |
|---|---|---|---|---|
| **Demographics** | | | | |
| Age median (IQR), y | 65.0 (56.0–72.0) | 71.0 (59.0–78.0) | 64.0 (54.0–69.0) | 0.001 |
| Sex | | | | 0.467 |
| Male | 109 (65.3%) | 38 (69.1%) | 71 (63.4%) | |
| Female | 58 (34.7%) | 17 (30.9%) | 41 (36.6%) | |
| **Laboratory findings** | | | | |
| Leucocytes (× $10^9$/L; normal range 3.5–9.5) | 7.6 (5.1–11.4) | 9.1 (6.2–13.4) | 6.8 (4.6–10.7) | 0.005 |
| Neutrophils (× $10^9$/ L; normal range 1.8–6.3) | 6.3 (3.8–10.2) | 7.9 (5.4–12.6) | 5.4 (3.4–9.6) | 0.004 |
| Lymphocytes (× $10^9$/L; normal range 1.1–3.2) | 0.6 (0.4–0.8) | 0.6 (0.5–0.7) | 0.6 (0.4–0.9) | 0.153 |
| D-dimer (μg/mL; normal range 0.0–0.5) | 2.5 (0.9–17.9) | 3.9 (1.2–21.0) | 1.8 (0.8–9.7) | 0.016 |
| Albumin (g/L; normal range 35.0–52.0) | 31.5 (27.9–35.0) | 29.8 (25.3–32.6) | 32.5 (29.5–35.3) | <0.001 |
| Lactate dehydrogenase (U/L; normal range 135.0–225.0) | 465.5 (338.0–597.5) | 504.0 (347.0–708.0) | 427.0 (325.0–529.0) | 0.007 |
| Glucose (mmol/L; normal range 3.9–6.1) | 7.7 (6.3–11.8) | 10.6 (7.4–16.9) | 6.9 (6.1–9.4) | <0.001 |
| Hypersensitive cardiac troponin (pg/mL; normal range 0.0–34.2) | 14.4 (4.7–121.9) | 41.5 (8.6–557.5) | 10.3 (3.8–59.9) | <0.001 |
| N-terminal pro-brain Natriuretic Peptide (pg/mL; normal range 0.0–247.0) | 504.0 (161.5–1895.0) | 790.0 (199.0–2636.0) | 353.5 (126.3–1546.0) | 0.032 |
| Procalcitonin (ng/mL; normal range 0.02–0.05) | 0.16 (0.07–0.59) | 0.23 (0.10–1.18) | 0.14 (0.05–0.37) | 0.005 |
| Serum ferritin (ug/L; normal range 30.0–400.0) | 1217.4 (613.2–2006.6) | 1715.8 (833.4–2429.3) | 1072.4 (538.9–1724.4) | 0.002 |
| Interleukin -1β (pg/mL; normal range 0.0–5.0)* | 5.0 (5.0–5.0) | 5.0 (5.0–5.0) | 5.0 (5.0–5.0) | 0.729 |
| Interleukin-2 receptor (U/mL; normal range 223–710) | 954.0 (643.5–1364.0) | 1083.0 (677.0–1501.0) | 923.5 (597.8–1311.3) | 0.053 |
| Interleukin-6 (pg/mL; normal range 0.0–7.0) | 36.3 (9.5–96.4) | 56.1 (24.8–145.3) | 25.9 (5.3–68.1) | 0.001 |
| Interleukin-8 (pg/mL; normal range 0.0–62.0) | 21.9 (10.5–47.3) | 28.3 (15.8–82.4) | 16.0 (9.5–36.3) | 0.001 |
| Interleukin-10 (pg/mL; normal range 0.0–9.1) | 9.0 (5.0–15.9) | 11.5 (7.9–19.4) | 6.7 (5.0–12.6) | 0.001 |
| Tumor Necrosis Factor α (pg/mL; normal range 0.0–8.1) | 9.4 (7.2–14.3) | 11.1 (7.7–20.6) | 9.1 (6.8–13.2) | 0.014 |
| **Treatments** | | | | |
| Antibiotic therapy | 150 (89.8%) | 54 (98.2%) | 96 (85.7%) | 0.012 |
| Antiviral therapy | 167 (100%) | 55 (100.0%) | 112 (100.0%) | - |
| Glucocorticoids | 135 (80.8%) | 48 (87.3%) | 87 (77.7%) | 0.191 |
| Non-IMV | 167 (100.0%) | 55 (100.0%) | 112 (100.0%) | - |
| IMV | 92 (55.1%) | 32 (58.2%) | 60 (53.6%) | 0.573 |
| ECMO | 4 (2.4%) | 1 (1.8%) | 3 (2.7%) | 1.000 |
| CRRT | 23 (13.8%) | 4 (7.3%) | 19 (17.0%) | 0.088 |
| **Outcomes** | | | | |
| Survival | 58 (34.7%) | 11 (20.0%) | 47 (42.0%) | 0.005 |
| Died | 109 (65.3%) | 44 (80.0%) | 65 (58.0%) | |

Data are median (IQR). p values were calculated by Mann-Whitney U test, $\chi^2$ test, or Fisher's exact test, as appropriate. Non-IMV: any oxygen therapy other than IMV; IMV: Invasive mechanical ventilation. ECMO: Extracorporeal Membrane Oxygenation; CRRT: continuous renal replacement therapy.

*Patients with normal level of interleukin-1β (lower than 5 pg/mL) were presented as "5 pg/mL".

PCT, hsCRP, IL-6, IL-10, TNFα and poorly controlled hyperglycemia were also associated with death. We included meaningful indicators from the univariate analysis of 55 diabetic patients in the multivariate Cox regression model. We found that lymphocyte count >0.45×$10^9$/L was a protective factor associated with survival in diabetic patients. LDH >600 U/L, hsCRP >90 mg/L and IL-10 >10 U/mL increased mortality (Table 3).

**Table 2. Demographic and laboratory findings in patients with COVID-2019 on admission.**

| Characteristics | Diabetic patient (n = 55) | | | Non-diabetic patient (n = 112) | | |
|---|---|---|---|---|---|---|
| | Non-survivor (n = 44) | Survivor (n = 11) | P Value | Non-survivor (n = 65) | Survivor (n = 47) | P Value |
| **Demographics** | | | | | | |
| Age median (IQR), y | 72.0 (63.0–78.0) | 57.0 (54.0–72.0) | 0.016 | 66.0 (61.0–73.0) | 55.0 (46.0–65.0) | <0.001 |
| Sex | | | 0.722 | | | 0.752 |
| Male | 31 (70.5%) | 7 (63.6%) | | 42 (64.6%) | 29 (61.7%) | |
| Female | 13 (29.5%) | 4 (36.4%) | | 23 (35.4%) | 18 (38.3%) | |
| **Poorly controlled hyperglycemia** | 39 (88.6%) | 6 (45.5%) | 0.019 | - | - | - |
| **Laboratory findings** | | | | | | |
| Leucocytes (×10$^9$/L; normal range 3.5–9.5) | 10.1 (7.9–14.7) | 5.6 (3.3–6.8) | <0.001 | 9.4 (5.5–12.8) | 5.2 (3.9–6.9) | <0.001 |
| Neutrophils (×10$^9$/L; normal range 1.8–6.3) | 9.5 (6.9–13.4) | 4.2 (2.6–5.7) | <0.001 | 8.1 (4.5–11.5) | 3.8 (2.7–5.6) | <0.001 |
| Lymphocytes (×10$^9$/L; normal range 1.1–3.2) | 0.5 (0.4–0.7) | 0.7 (0.7–0.9) | 0.006 | 0.5 (0.4–0.7) | 0.8 (0.6–1.0) | <0.001 |
| D-dimer (μg/mL; normal range 0.0–0.5) | 8.9 (2.3–21.0) | 1.0 (0.4–1.5) | <0.001 | 4.3 (1.6–21.0) | 0.7 (0.4–1.6) | <0.001 |
| Albumin (g/L; normal range 35.0–52.0) | 29.4 (24.6–31.7) | 30.9 (27.8–35.7) | 0.058 | 31.0 (28.7–34.2) | 35.0 (31.9–39.4) | <0.001 |
| Lactate dehydrogenase (U/L; normal range 135.0–225.0) | 547.5 (444.8–772.5) | 259.0 (243.0–479.0) | <0.001 | 479.0 (384.3–608.3) | 347.0 (233.0–419.0) | <0.001 |
| Glucose (mmol/L; normal range 3.9–6.1) | 11.3 (7.7–17.7) | 8.6 (7.0–13.5) | 0.265 | 7.7 (6.6–10.9) | 6.5 (5.7–7.3) | <0.001 |
| Hyper-sensitive cardiac troponin (pg/mL; normal range 0.0–34.2) | 71.4 (18.7–1072.2) | 5.2 (2.3–8.5) | <0.001 | 31.2 (12.1–159.3) | 3.8 (2.4–5.9) | <0.001 |
| N-terminal pro-brain Natriuretic Peptide (pg/mL; normal range 0.0–247.0) | 1603.0 (424.5–3654.50) | 175.5 (66.5–642.8) | 0.001 | 913.5 (333.5–3644.8) | 120.0 (83.8–323.5) | <0.001 |
| Procalcitonin (ng/mL; normal range 0.02–0.05) | 0.39 (0.14–1.33) | 0.06 (0.04–0.59) | 0.006 | 0.18 (0.09–0.64) | 0.06 (0.04–0.18) | <0.001 |
| Serum ferritin (ug/L; normal range 30.0–400.0) | 1814.7 (1034.1–2722.3) | 781.7 (582.7–1964.0) | 0.044 | 1351.0 (915.2–2429.2) | 662.7 (438.8–1270.2) | <0.001 |
| Interleukin -1β(pg/mL; normal range 0.0–5.0)* | 5.0 (5.0–5.0) | 5.0 (5.0–5.0) | 0.844 | 5 (5–5) | 5 (5–5) | 0.263 |
| Interleukin-2 receptor (U/mL; normal range 223–710) | 1163.0 (946.5–1586.7) | 677.0 (595.0–1035.0) | 0.023 | 1128.0 (762.0–1809.0) | 694.0 (459.0–929.0) | <0.001 |
| Interleukin-6(pg/mL; normal range 0.0–7.0) | 68.0 (35.6–220.4) | 22.2 (2.2–90.0) | 0.006 | 57.9 (16.8–130.9) | 5.8 (2.3–30.3) | <0.001 |
| Interleukin-8(pg/mL; normal range 0.0–62.0) | 29.4 (19.5–104.1) | 21.8 (10.6–64.5) | 0.169 | 26.6 (12.4–77.0) | 10.0 (7.3–19.5) | <0.001 |
| Interleukin-10(pg/mL; normal range 0.0–9.1) | 13.5 (9.3–19.9) | 7.8 (5.0–11.9) | 0.023 | 10.8 (5.7–20.7) | 5.0 (5.0–8.3) | <0.001 |
| Tumor Necrosis Factor α (pg/mL; normal range 0.0–8.1) | 11.8 (7.7–21.6) | 9.7 (7.7–11.6) | 0.236 | 9.9 (7.2–15.0) | 8.3 (6.1–10.8) | 0.045 |

Data are median (IQR). p values were calculated by Mann-Whitney U test.

*Patients with normal level of interleukin-1β (lower than 5 pg/mL) were presented as "5 pg/mL".

Diabetic patients had higher mortality rate than nondiabetic patients (80.0% vs 58.0%, P = 0.005) (Table 2). There was a significant difference in overall survival (OS) between diabetic patients and nondiabetic patients (P<0.001) (Fig 2 and S5 Table). This finding suggests that diabetes is a risk factor for reduced survival in severe cases. In diabetic patients, patients with poorly controlled hyperglycemia had shorter OS than patients with well controlled hyperglycemia (Fig 2 and S5 Table). According to the multivariate cox regression model, four laboratory results affecting the survival of diabetic patients were calculated (lymphocyte count ≤0.45× 10$^9$/L, CRP >90 mg/L, LDH >600 U/L, IL-10 >10 U/mL) (Table 3). The diabetic patients were divided into 5 groups according to the number of abnormal items, with significant differences in OS in the 5 groups (P<0.001) (Fig 2 and S5 Table).

**Table 3. Cox regression analysis of risk factors for mortality of diabetic patients with COVID-19.**

| Characteristics | Univariable HR (95% CI) | P value | Multivariable HR (95% CI) | P value |
|---|---|---|---|---|
| Age, years | | | | |
| ≤70 | 1 (ref) | | | |
| >70 | 1.891 (1.034–3.456) | 0.039 | 1.427 (0.516–3.951) | 0.493 |
| Neutrophils (× 10⁹/L) | | | | |
| ≤7 | 1 (ref) | | | |
| >7 | 3.194 (1.586–6.429) | 0.001 | 0.447 (0.096–2.074) | 0.304 |
| Lymphocytes (× 10⁹/L) | | | | |
| ≤0.45 | 1 (ref) | | | |
| >0.45 | 0.316 (0.161–0.621) | 0.001 | 0.142 (0.026–0.787 | 0.025 |
| Platelets (× 10⁹/L) | | | | |
| ≤65 | 1 (ref) | | | |
| >65 | 0.351 (0.152–0.815) | 0.015 | 2.915 (0.561–15.161) | 0.203 |
| Lactate dehydrogenase (U/L) | | | | |
| ≤600 | 1 (ref) | | | |
| >600 | 2.919 (1.542–5.527) | 0.001 | 10.340 (1.647–64.904) | 0.013 |
| Serum creatinine (μmol/L) | | | | |
| ≤85 | 1 (ref) | | | |
| >85 | 3.082 (1.667–5.701) | <0.001 | 1.826 (0.416–8.011) | 0.425 |
| BUN (mmol/L) | | | | |
| ≤10 | 1 (ref) | | | |
| >10 | 5.674 (2.791–11.536) | <0.001 | 0.577 (0.075–4.461) | 0.598 |
| Hypersensitive cardiac troponin (pg/mL) | | | | |
| ≤100 | 1 (ref) | | | |
| >100 | 2.286 (1.243–4.206) | 0.008 | 1.180 (0.344–4.054) | 0.792 |
| N-terminal pro-brain Natriuretic Peptide (pg/mL) | | | | |
| ≤1500 | 1 (ref) | | | |
| >1500 | 3.379 (1.757–6.498) | <0.001 | 1.889 (0.444–8.039) | 0.389 |
| Procalcitonin (ng/mL) | | | | |
| ≤0.15 | 1 (ref) | | | |
| >0.15 | 2.333 (1.202–4.527) | 0.012 | 0.432 (0.113–1.654) | 0.220 |
| Hypersensitive C-reactive protein I (mg/L) | | | | |
| ≤90 | 1 (ref) | | | |
| >90 | 2.200 (1.199–4.038) | 0.011 | 5.459 (1.532–19.451) | 0.009 |
| Interleukin-2 receptor (U/mL) | | | | |
| ≤1000 | 1 (ref) | | | |
| >1000 | 1.735 (0.927–3.246) | 0.085 | | |
| Interleukin-6 (U/mL) | | | | |
| ≤120 | 1 (ref) | | | |
| >120 | 2.313 (1.185–4.514) | 0.014 | 1.135 (0.230–5.598) | 0.877 |
| Interleukin-10 (U/mL) | | | | |
| ≤10 | 1 (ref) | | | |
| >10 | 3.088 (1.489–6.403) | 0.002 | 6.868 (1.291–36.550) | 0.024 |
| Tumor Necrosis Factor α (pg/mL) | | | | |
| ≤20 | 1 (ref) | | | |
| >20 | 2.198 (1.083–4.461) | 0.029 | 0.319 (0.063–1.604) | 0.166 |
| Control of hyperglycemia | | | | |

(*Continued*)

**Table 3.** (Continued)

| Characteristics | Univariable | P value | Multivariable | P value |
|---|---|---|---|---|
| | HR (95% CI) | | HR (95% CI) | |
| Poorly controlled | 2.771 (1.083–7.087) | 0.033 | 0.629 (0.149–2.633) | 0.529 |
| Well controlled | 1 (ref) | | | |

HR = hazard ratio.

## Discussion

In this single-center, retrospective and observational study, we enrolled 167 patients with severe COVID-19, with a median age of 65.0 years (IQR 56.0–72.0). Through the comparison of survivors (65.27%) and nonsurvivors (34.73%), it was found that there was a significant difference in the proportion of patients with endocrine diseases and digestive system diseases. Through further COX regression, univariate and multivariate analyses showed that older age, diabetes, neutrophil count, lymphocyte count and blood glucose were independent prognostic factors in patients with COVID-19. Diabetes was the only concomitant disease that affected prognosis.

Individuals with diabetes are at risk of severe infections, especially influenza and pneumonia [9]. Diabetes was an important risk factor for exacerbation or death after infection in Middle East Respiratory Syndrome Coronavirus (MERS-CoV) studies [10–12]. In a severe acute respiratory syndrome (SARS) study, diabetes history and hyperglycemia were found to be independent predictors of death and morbidity [13]. At present, the researches on the prognosis of patients with diabetes and SARS-CoV-2 is limited and contradictory. In a study of 140 patients with COVID-19, 58.5% of the patients were nonsevere and only 12.1% of which had diabetes, suggesting that diabetes had no adverse effects in patients with mild disease [14]. However, some studies showed that type 2 diabetes increased severity of COVID-19 and it was an important risk factor for COVID-19 progression and adverse endpoints [15, 16]. What's more, a report on 72,314 COVID-19 cases analyzed by the Chinese Centre for Disease Control and Prevention showed that the mortality rate of people with diabetes is increased [9]. The results of our study also suggested that diabetes was an independent prognostic factor for COVID-19 patients. In our study, the survival rate of diabetic patients was lower, and the time from attack to death was shorter than nondiabetic patients. Besides, in the study of SARS, it was found that viruses could enter the islet through combining to angiotensin converting enzyme 2 (ACE2), damage the pancreatic β-cells and cause acute diabetes [17]. ACE2 was expressed in various tissues and organs of the human body. Significant ACE2/Ang1-7/Mas signaling pathway inhibition and increased ACE/AngII/AT1R pathway activity occur after SARS-CoV-2 bind to ACE2 in vivo [18]. The mortality occurs through the imbalance of the renin-angiotensin system (RAS) and an increase in the level of inflammatory factors [18]. It was shown that ACE2 was also the receptor of SARS-CoV-2 [14, 19, 20], therefore SARS-CoV-2 may also directly damage the pancreas through ACE2. According to the comparative analysis of survivors and nonsurvivors, it was found that in nondiabetic patients, the blood glucose of nonsurvivors was also significantly higher than survivors, indicating that even if there was no history of diabetes, hyperglycemia was also an adverse prognostic factor.

Through further research, compared with nondiabetic patients, levels of cardiac injury indexes (cardiac troponin, BNP), inflammation related indexes (PCT, ferritin), coagulation function (d-dimer), cytokines, and LDH were markedly elevated in diabetic patients. These were consistent with the findings of Lihua et al. [16]. Besides, albumin decreased significantly

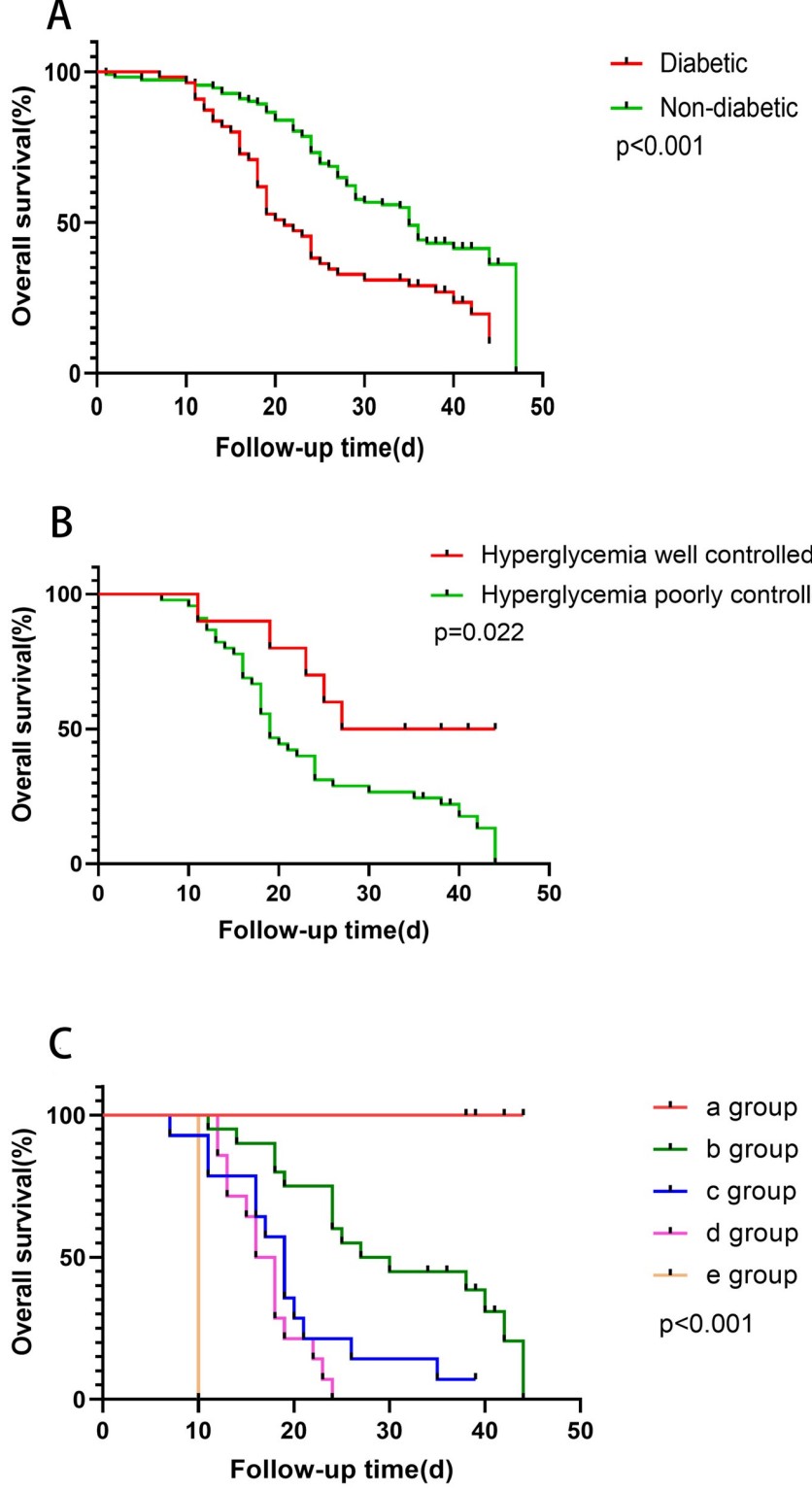

**Fig 2. Overall Survival (OS) of severe patients with COVID-19.** The y-axis represents the OS rate. Data were analyzed by the Kaplan-Meier method. (A) The OS of diabetic patients and nondiabetic patients ($\chi2 = 14.405$, P<0.001). (B) Diabetic patients with poorly controlled hyperglycemia had shorter OS than patients with well controlled hyperglycemia (P = 0.022). (C) The OS of five groups by abnormal laboratory indicators in different degrees ($\chi2 = 58.62$, P<0.001). When laboratory indicators exceed the exceptions listed below, they are included in our groups,

including lymphocyte count $\leq 0.45 \times 10^9$/L, CRP > 90 mg/L, LDH > 600 U/L, IL-10 >10 U/mL. In group a, none of the above four indicators were abnormal; in group b, one indicator was abnormal; in group c, two indicators were abnormal; in group d, three indicators were abnormal; and in group e, four indicators were abnormal.

in diabetes, indicating that the basic condition of diabetic patients was poor. We further analyzed the prognosis of patients with diabetes and determined their cutoff points to make these prognostic factors specific, and then verified these factors through Kaplan-Meier method. Eventually, the result showed that lymphocyte count $\leq 0.45 \times 10^9$/L, LDH >600 U/L, hsCRP >90 mg/L and IL-10 >10 U/mL were associated with increased risk of death. Improved glycemic control was associated with better outcomes in patients with COVID-19 and pre-existing diabetes [16]. In this study, well controlled hyperglycemia was negatively correlated with mortality and OS in patients with diabetes, however, this correlation was mitigated in multivariate analysis.

Lymphocytopenia is a common characteristic in COVID-19 patients [21], and lymphocyte count $\leq 0.45 \times 10^9$/L was an independent prognostic factor for patients with diabetes in this study. Researches have shown that the immune cells of people and mice with type 2 diabetes had a transition from regulatory T cells (Treg) or anti-inflammatory macrophages to proinflammatory macrophages and Th1 and Th17CD4+T cells [22, 23]. The increase of LDH and CRP indicated that the injury of the body was severe and may be complicated with bacterial infection which affected the prognosis.

Acute lung injury (ALI) is a common consequence of a cytokine storm in the lung alveolar environment and circulatory system and is most commonly associated with suspected or proven infections in the lungs or other organs [24]. Pathogen-induced lung injury can progress into ALI or its more severe form, ARDS, as seen with SARS and influenza virus infections. In patients with SARS, pro-inflammatory factors (IL-1β, IL-6, IL-12, interferon-γ, inducible protein-10, monocyte chemoattractant protein-1) increased [22]. Cytokine storms are known to play an important role in the acute lung injury of patients with severe COVID-19 [25]. The IL-6 and IL-8 levels were markedly increased, consistent with the severity of the disease. However, compared with the survivors, IL-10 also increased significantly in our study, and the mortality of patients with severe COVID-19 complicated with diabetes was increased when IL-10 >10 U/mL. The relation between IL-10 and diabetes needs further research. Moreover, the level of cytokines in patients with diabetes was higher than nondiabetes, suggesting that diabetic patients were more likely to experience cytokine storms, thus aggravating the lung damage. Twenty-three patients (including four patients with diabetes) with COVID-19 received continuous renal replacement therapy (CRRT) treatment but eventually died, indicating that it may be too late to use CRRT at these patients.

In this study, the age of the nonsurvivors was significantly higher than survivors in both diabetic and nondiabetic patients, and the univariate and multivariate results were consistent. As age increases, the underlying inflammation and immune system disorders associated with aging may be related to the reactivation of latent viral infections and the release of endogenous damage-related pattern recognition receptor (PRR) ligands [26]. Additionally, there are more complications in the elderly [15]. Therefore, we should be particularly vigilant against the progress of the disease and implement the necessary treatment promptly for elderly patients.

133 (79.6%) patients received glucocorticoid treatment in this study, of which 47 cases were complicated with diabetes. Inflammation and cytokine storms accelerate the progression of the disease in patients with severe disease symptoms. It is recommended that glucocorticoids could be used for a short period time (3–5 days) according to the degree of dyspnea and chest imaging progress. The recommended dose could not exceed the equivalent

methylprednisolone dosage of 1-2mg/kg/d [2, 27, 28], monitoring the blood glucose and controlling by insulin at the same time. In this study, the use of glucocorticoids in patients with diabetes did not affect the prognosis.

The present study has some limitations. First, the number of patients in the present study was limited, and we need a larger sample to verify our conclusion. Second, due to the retrospective analysis, there was the problem of incomplete case information. Finally, the present study included only severe cases and lacked comparisons with mild patients.

## Conclusions

This study showed that diabetes, age, neutrophil count and lymphocyte count, blood glucose were independent prognostic factors in patients with COVID-19, but lack of materialization for clinical application. Through further analysis of the prognosis in diabetic patients with severe COVID-19, it was found that lymphocyte count $\leq 0.45 \times 10^9$/L, LDH >600 U/L, hsCRP >90 mg/L and IL-10 >10 U/mL were associated with poor prognosis. It is more valuable to judge the condition through the combination of two or more prognostic factors above. People with these characteristics should be taken seriously. Cutoff points were determined and verified to make these prognostic factors specific and increase prognostic value.

## Supporting information

**S1 Table. Demographics, clinical characteristics and treatment of patients.**
(DOCX)

**S2 Table. Cox regression analysis of risk factors for mortality among severe patients.**
(DOCX)

**S3 Table. Laboratory results of patients with corona virus disease 2019.**
(DOCX)

**S4 Table. Laboratory results of patients with corona virus disease 2019 on admission.**
(DOCX)

**S5 Table. Overall survival of patients (diabetic patients and nondiabetic patients, diabetic patients affected with glycemia control, groups according to the number of abnormal items).**
(XLSX)

## Acknowledgments

We thank all health-care workers involved in the diagnosis and treatment of patients in Wuhan.

## Author Contributions

**Conceptualization:** Donghua Zhang.

**Data curation:** Zhiqiong Wang, Lifang Huang.

**Formal analysis:** Xia Mao.

**Writing – original draft:** Yan Hui, Yi Li, Xiwen Tong.

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
