## [Decision Letter · Decision Letter 0]

8 Oct 2020

PONE-D-20-21885

The risk factors for mortality of diabetic patients with severe COVID-19: a retrospective study of 167 severe COVID-19 cases in Wuhan

PLOS ONE

Dear Dr. Zhang,

Thank you for submitting your manuscript to PLOS ONE. After careful consideration, we feel that it has merit but does not fully meet PLOS ONE’s publication criteria as it currently stands. Therefore, we invite you to submit a revised version of the manuscript that addresses the points raised during the review process.

Please see comments by the reviewers.  In you revised manuscript, kindly provide point by point response to their queries.

We look forward to receiving your revised manuscript.

Kind regards,

Muhammad Adrish

Academic Editor

PLOS ONE

Journal Requirements:

2. Please include the date(s) on which you accessed the databases or records to obtain the data used in your study.

'The funders had no role in study design, data collection and analysis, decision to publish, or preparation of the manuscript.'

*Please include your amended statements within your cover letter; we will change the online submission form on your behalf.*

Reviewers' comments:

Reviewer's Responses to Questions

**Comments to the Author**

1. Is the manuscript technically sound, and do the data support the conclusions?

Reviewer #1: Yes

Reviewer #2: Yes

2. Has the statistical analysis been performed appropriately and rigorously? 

Reviewer #1: Yes

Reviewer #2: Yes

3. Have the authors made all data underlying the findings in their manuscript fully available?

Reviewer #1: Yes

Reviewer #2: No

4. Is the manuscript presented in an intelligible fashion and written in standard English?

Reviewer #1: Yes

Reviewer #2: Yes

5. Review Comments to the Author

Reviewer #1: The manuscript is well written and technically sound with good statistical analysis and would make it's contribution to the ongoing battle against COVID-19. However, it requires few correction before it can be accepted for publication. There are few typo and grammatical errors and the discussion should be modified with the two articles that I recommend by comparing them with your results. See the corrections in the PDF attachment.

Reviewer #2: In this article, the authors reviewed the records of 167 patients with severe COVID-19 who were hospitalized at Tongji Hospital in Wuhan China between late January and early March of 2020. Of these patients they identified 55 who had diabetes and 112 who did not.

Baseline demographics and admission laboratory data were compared between the patients with and without diabetes which showed significant differences in multiple categories including age, wehigth blood cell counts, albumin and multiple inflammatory and, particularly, cardiac markers.

Mortality rates were significantly higher in patients with diabetes (80% vs 58%). In a univariate analysis comparing patients who did and did not survive COVID-19 multiple markers were disproportionately elevated in both those with and without diabetes. These meaningful indicators were included in a multivariate COX regression model which determined cutoffs for lymphocyte counts, LDH, hsCRP, and IL-10 which were predictive of survival/mortality. Patients who had more than one abnormality had higher mortality rates.

This manuscript provides novel data regarding inflammatory markers in patient with severe COVID-19 and while the overall trends in markers are similar between those with and without diabetes the multivariate cutoffs could be useful comparisons for future research.

Some additional clarity regarding the methods and data would provide readers more context.

Were the 167 patients with severe COVID-19 all of the patients with that condition seen between January and March or were some excluded? On page 6 it is mentioned that “we included 167 patients with complete data…” If some patients were excluded due to incomplete data, the methods of how this was decided should be outlined.

Additionally, the specific criteria used to classify patients as “severe” COVID-19 should be outlined as reference 4 is in Chinese and would not be readable by most of the readership of PLOS One. If these criteria are different than what is currently used as an international standard these differences and their implications should be summarized as well.

Figure 1 shows differences between laboratory values at admission and discharge/death, but these data at the time of discharge/death do not appear to be presented elsewhere in tables or analysis. The study otherwise appears focused on laboratory values on admission. Was a comparison of these makers performed between patients with and without diabetes who died/survived?

Was any analysis performed regarding the degree of glycemic control during admission for patients with diabetes? This would be a very clinically relevant comparison and of interest to readers. If the focus is risk determined solely by laboratory values on admission, that is a reasonable analysis, but it should be more clearly stated.

The values for interleukin-1beta in the tables are all listed as 5, I suspect this was entered in error.

6. PLOS authors have the option to publish the peer review history of their article (what does this mean?). If published, this will include your full peer review and any attached files.

Reviewer #1: **Yes: **Adekunle Sanyaolu

Reviewer #2: No

---

## [Author Response · Author response to Decision Letter 0]

30 Oct 2020

Author’s Response to the Editor and Reviewers

Title: The risk factors for mortality of diabetic patients with severe COVID-19: a retrospective study of 167 severe COVID-19 cases in Wuhan

ID: PONE-D-20-21885

Version: 1

Date: October 21th, 2020

Corresponding authors: Donghua Zhang (zdh_62@126.com) 

Dear Editor,

We sincerely appreciate all the comments and suggestions of the reviewers and editor to strengthen the manuscript and felt encouraged by your positive feedback. We have extensively revised our paper according to those comments and suggestions, and the changes with track changes in word program are in our revised manuscript. We hope that our efforts have adequately addressed your concerns. Below, we have provided a point-by-point response to the comments. 

We look forward to your response.

Sincerely yours,

Donghua Zhang, MD, Ph.D 

Department of Hematology, Tongji Hospital

Tongji Medical College

Huazhong University of Science and Technology 

1095 Jie-Fang Avenue, Wuhan

Hubei, 430030, P. R. China

Phone: 0086-27-83662830

Point-by-Point Response to the Reviewers

Reviewer 1: 

1. Response to comment: There are few typo and grammatical errors.

Response: We appreciate the suggestion of the reviewer. We have revised those typo and grammatical errors for better understanding according to the reviewer. 

2. Response to comment: The discussion should be modified with the two articles that I recommend by comparing them with your results. See the corrections in the PDF attachment. 

Response: We sincerely appreciate the reviewer's comment. We have carefully read the two articles you have recommend and we modified the discussion by comparing the two articles with our results. We sincerely thank you for the recommendation and corrections in the PDF attachment, which helped to improve the manuscript greatly. Thank you again.

Reviewer 2

1. Response to comment: Were the 167 patients with severe COVID-19 all of the patients with that condition seen between January and March or were some excluded? On page 6 it is mentioned that “we included 167 patients with complete data…” If some patients were excluded due to incomplete data, the methods of how this was decided should be outlined. 

Response: We sincerely appreciate the reviewer's comment. All of the patients with severe COVID-19 with that condition were seen between January and March in our hospital, and no patients were excluded. On page 6, "we included 167 patients with complete data...". We mentioned this sentence to emphasize that our data are complete. In fact, information of all the inpatients were recorded in detail, and none of them was excluded because the complete data. This sentence might cause misunderstanding, we have modified it for a better understanding. Thank you.

2. Response to comment: Additionally, the specific criteria used to classify patients as “severe” COVID-19 should be outlined as reference 4 is in Chinese and would not be readable by most of the readership of PLOS One. If these criteria are different than what is currently used as an international standard these differences and their implications should be summarized as well. 

Response: We sincerely appreciate the reviewer's comment. The diagnostic criteria mentioned in reference 4 was the only criteria for diagnostic classification in China during early stage of the pandemic and was also cited by some published articles (Yin L, et al. PLoS ONE, 2020; Fei Z, et al. Lancet, 2020). Internationally, the criteria of ARDS and the diagnostic criteria of sepsis and septic shock can be used to define severe COVID-19 patients (Cascella M, et al. Features, Evaluation, and Treatment of Coronavirus (COVID-19). In: StatPearls. 2020). It requires clinical and ventilatory criteria which is suggestive of a serious new-onset respiratory failure and different degrees of hypoxia. Actually, respiratory failure and hypoxia is just the point that was focused on by the criteria in our manuscript. Therefore, the criteria cited in our manuscript are consistent with international standard. For the convenience of understanding by most of the readership of PLOS One, we have translated the diagnostic criteria into English and cited the criteria of ARDS in the Method part of the revised manuscript. 

3. Response to comment: Figure 1 shows differences between laboratory values at admission and discharge/death, but these data at the time of discharge/death do not appear to be presented elsewhere in tables or analysis. The study otherwise appears focused on laboratory values on admission. Was a comparison of these makers performed between patients with and without diabetes who died/survived?

Response: We sincerely appreciate the reviewer's comment. The data in Figure 1 has been presented in S3 Table as supplementary material. A comparison of these markers performed between patients with and without diabetes who died/survived has been showed in S4 Table as supplementary material. Thanks a lot for your advices.

4. Response to comment: Was any analysis performed regarding the degree of glycemic control during admission for patients with diabetes? This would be a very clinically relevant comparison and of interest to readers. If the focus is risk determined solely by laboratory values on admission, that is a reasonable analysis, but it should be more clearly stated.

Response: We appreciate the reviewer for the suggestion. This is an important question. We have reviewed data regarding the degree of glycemic control during admission for patients with diabetes from the electronic medical records. The definition of poorly controlled and well controlled hyperglycemia was based on the published criteria (Bruce et al. Glycemic Characteristics and Clinical Outcomes of COVID-19 Patients Hospitalized in the United States, Journal of Diabetes Science and Technology, 2020). We found that the poorly controlled hyperglycemia was associated with poor prognosis in diabetic patients in univariate regression analysis, but showed no significance in multivariate analysis. Moreover, diabetic patients with poorly controlled hyperglycemia had shorter OS than patients with well controlled hyperglycemia. We added these data in Table 1-3 and added a new figure as Fig 2B in in revised manuscript. Therefore, risks to poor prognosis in severe patients with COVID-19 were mainly determined by laboratory values on admission according to our conclusion in this manuscript. 

5. Response to comment: The values for interleukin-1beta in the tables are all listed as 5, I suspect this was entered in error.

Response: We sincerely appreciate the reviewer's carefulness. In our hospital, if the patient's interleukin-1beta level is lower than 5 pg/mL (normal range: 0.0–5.0 pg/mL), it will present as “<5 pg/mL”, which just shows normal values. Actually, the values for interleukin-1beta in the tables are all listed as 5 because interleukin-1beta of most patients were normal. In the table, median and IQR were used to represent this continuous variable, so it was shown as 5.0 (5.0-5.0). Thank you very much for raising this question. We will make an explanation at the bottom of the table for readers' understanding.

---

## [Decision Letter · Decision Letter 1]

25 Nov 2020

The risk factors for mortality of diabetic patients with severe COVID-19: a retrospective study of 167 severe COVID-19 cases in Wuhan

PONE-D-20-21885R1

Dear Dr. Zhang,

We’re pleased to inform you that your manuscript has been judged scientifically suitable for publication and will be formally accepted for publication once it meets all outstanding technical requirements.

Kind regards,

Muhammad Adrish

Academic Editor

PLOS ONE

Additional Editor Comments (optional):

Reviewers' comments:

Reviewer's Responses to Questions

**Comments to the Author**

1. If the authors have adequately addressed your comments raised in a previous round of review and you feel that this manuscript is now acceptable for publication, you may indicate that here to bypass the “Comments to the Author” section, enter your conflict of interest statement in the “Confidential to Editor” section, and submit your "Accept" recommendation.

Reviewer #1: All comments have been addressed

Reviewer #2: All comments have been addressed

2. Is the manuscript technically sound, and do the data support the conclusions?

Reviewer #1: Yes

Reviewer #2: Yes

3. Has the statistical analysis been performed appropriately and rigorously? 

Reviewer #1: Yes

Reviewer #2: Yes

4. Have the authors made all data underlying the findings in their manuscript fully available?

Reviewer #1: Yes

Reviewer #2: Yes

5. Is the manuscript presented in an intelligible fashion and written in standard English?

Reviewer #1: Yes

Reviewer #2: Yes

6. Review Comments to the Author

Reviewer #1: There is a typo correction for you to fix.

Discussion, Page 15: Place the comma after '2' in the number 7,2314 to read 72,314.

Reviewer #2: The author's clarifications have answered all of my questions. I think this does provide usefull data regarding the correlation between inflammatory makers and outcomes in COVID-19 from early in the pandemic and should be published.

7. PLOS authors have the option to publish the peer review history of their article (what does this mean?). If published, this will include your full peer review and any attached files.

Reviewer #1: No

Reviewer #2: No

---

## [Editor Report · Acceptance letter]

21 Dec 2020

PONE-D-20-21885R1 

The risk factors for mortality of diabetic patients with severe COVID-19: a retrospective study of 167 severe COVID-19 cases in Wuhan 

Dear Dr. Zhang:

I'm pleased to inform you that your manuscript has been deemed suitable for publication in PLOS ONE. Congratulations! Your manuscript is now with our production department. 

Kind regards, 

on behalf of

Dr. Muhammad Adrish 

Academic Editor

PLOS ONE